# Could Dental Material Reuse Play a Significant Role in Preservation of Raw Materials, Water, Energy, and Costs? Microbiological Analysis of New versus Reused Healing Abutments: An In Vitro Study

**DOI:** 10.3390/bioengineering11040387

**Published:** 2024-04-16

**Authors:** Roberto Burioni, Lucia Silvestrini, Bianca D’Orto, Giulia Tetè, Matteo Nagni, Elisabetta Polizzi, Enrico Felice Gherlone

**Affiliations:** 1Department of Microbiology and Virology, University “Vita-Salute” San Raffaele, 20132 Milan, Italy; burioni.roberto@hsr.it (R.B.); lucia.silvestrini@dgforlife.it (L.S.); 2Department of Dentistry, Dental School, IRCCS San Raffaele Hospital, “Vita-Salute” San Raffaele University, 20132 Milan, Italy; dortobianca21@gmail.com (B.D.); tetegiulia92@gmail.com (G.T.); nagnimatteo@hotmail.it (M.N.); 3Chair Center for Oral Hygiene and Prevention, Department of Dentistry, Dental School, IRCCS San Raffaele Hospital, “Vita-Salute” San Raffaele University, 20132 Milan, Italy; 4Dental School, IRCCS San Raffaele Hospital, “Vita-Salute” University, 20132 Milan, Italy; gherlone.enrico@hsr.it

**Keywords:** healing abutment, contamination, dental implant, sterilization, dental disinfectants

## Abstract

Aim: The objective of this in vitro study was to compare reused and sterilized versus new healing abutments to assess whether a decontamination and sterilization process performed on resued healing abutments was sufficient to remove residual proteins. The two groups were comparable with respect to patient safety. Materials and methods: During the period from September 2022 to October 2023, healing abutment screws were selected and divided into two groups according to whether they were new or previously used in patients. The samples were subjected to a decontamination and sterilization protocol, and results from sample sterility evaluation and assessment of surface protein levels were recorded. Results: The obtained results revealed a significant difference in the OD562 nm values between new and reused healing abutment samples. The assay demonstrates how treated healing abutments were still contaminated by residual proteins. Conclusion: Within the limitations of the present study, although from an infectious point of view sterilization results in the total eradication of pathogens, surface proteins remain on reused healing abutments.

## 1. Introduction

Dental implants not eligible for immediate loading are covered for the entire osseointegration period; then, a healing screw (also known as a healing abutment or healing cap) is often temporarily placed onto the implant to promote the creation of a peri-implant mucosal barrier, consisting of connective tissue and junctional epithelium, to provide an ideal lodging site for the final prosthetic restoration and to create a protective seal to prevent infection of the implant site [1,2,3].

Manufacturers widely acknowledge healing abutments (HAs) as essential disposable components within the realm of dental implant procedures [4,5].

However, reusing healing screws not only promotes saving valuable materials, such as titanium, but also contributes to cost reduction, pollution reduction, and streamlined production, supporting the World Health Organization’s goals for 2030, which aim to promote sustainable development in the healthcare sector through eco-friendly and responsible practices [6,7,8].

The main reason for considering abutments as disposable and non-reusable could be their intrinsic susceptibility to contamination and thread wear. These factors, if left unaddressed, have the potential to induce harm to not only the internal structure of the implant but also the engagement area of the screw, and even the overall composition of the titanium material [9,10,11]. Nevertheless, it is noteworthy that the practice of utilizing sterilized HAs across multiple procedures is commonplace [4].

When considering analogous practices within the field of dentistry, it is apparent that stainless steel burs, titanium drills, and other reusable instruments are extensively employed. For instance, adhering to manufacturers’ stipulations, implant drills are regarded as reusable, with an established regimen that involves an ultrasonic bath to eliminate physical debris, followed by autoclaving to neutralize any potentially infectious agents [12].

One of the central apprehensions surrounding the reuse of healing screws pertains to the potential contamination these components could harbor post-use. The amalgamation of plaque biofilm, gingival epithelium, blood, and saliva can culminate in the presence of free proteins and amino acids that firmly attach to the titanium surface, resulting in arduous removal efforts. Consequently, the residual debris on reused healing abutments can disrupt the precise alignment between the abutment and the implant, inadvertently introducing contaminants into the surrounding soft tissue. This could lead to a cascade of effects, including the initiation of an immune and inflammatory response due to the presence of denatured proteins [13,14,15].

A parallel concern associated with the recurrent utilization of healing screws lies in the potential biological implications of undergoing multiple sterilization cycles. The prospect of surface alterations could potentially compromise the strength of the transmucosal barrier, inadvertently creating an environment conducive to bacterial infiltration. This unintended consequence has the potential to interfere with the overall success of the dental implant [4,16,17,18].

In summary, while manufacturers resolutely advocate for the disposability of healing abutments in dental implant procedures, the broader context of reusability within the field of dentistry cannot be ignored. The intricate interplay among factors such as contamination, surface alterations, and biological responses underscores the significance of judiciously considering the trade-offs and implications of either approach.

The aim of this in vitro study was to compare reused and sterilized healing abutments (TG = Test Group) versus new healing abutments (CG = Control Group) to investigate whether the decontamination and sterilization process applied in the TG was sufficient to remove both pathogenic infective agents and residual proteins. The final outcome was to assess whether reused and sterilized healing abutments could be considered as safe as new in cases of reuse.

## 2. Materials and Methods

The study was performed during the period from September 2022 to October 2023 at Dentistry and Microbiology Departments, IRCCS San Raffaele Hospital, Milan, Italy. The research was conducted in accordance with the tenets of the Declaration of Helsinki and followed the Strengthening the Reporting of Observational Studies in Epidemiology (STROBE) guidelines for cohort studies (http://www.strobe-statement.org, accessed on 24 April 2021).

### 2.1. Sample Selection

The healing screws analyzed in this study were manufactured by Winsix, Biosafin, Cefla, Imola, Italy.

Details of type and features of the healing screws used for the study are given in the table below (Table 1).

Healing screws were obtained from patients in the Test Group based on the following inclusion and exclusion criteria:**Inclusion criteria**Age over 18 years;Systemic health according to American Society of Anesthesiologists guidelines (ASA I or II) [19];Non-smokers [20];Undergoing single implant placement in the mandibular or maxillary molar region;Screwed prostheses;Implant diameter of 3.8 mm;Not adequate for immediate loading and therefore undergoing deferred prosthetic loading;Healing screw inserted that corresponds to the characteristics listed in Table 1;Absence of periodontal disease with probing depth ≤ 3 mm, negativity for bleeding on probing and adherent gingiva ≥ 4 mm.**Exclusion criteria**Smokers;Uncontrolled systemic diseases [21];Taking drugs known to affect periodontal tissues or host immunity (e.g., phenytoin and steroids) [22,23];Diabetes;On chronic antiplatelet/anticoagulant therapy in the preceding six months [24,25];Patients who had taken antibiotics in the month prior to healing abutment removal [26].

Only one healing screw was taken from each patient. Every healing abutment was taken at implant reopening, which was performed following the bone healing period (4/5 months for the upper jaw and 3 months for the lower jaw).

For the purpose of the study, 40 healing abutments in the TG and 35 in the CG were analyzed. The steps to which the screws in both groups were exposed are outlined as follows (Figure 1).

### 2.2. Decontamination and Sterilization Protocol

The protocol involved the following steps:Decontamination [27,28,29]; HAs are soaked for 15 min in decontamination baths containing 2 g of powdered disinfectant solution diluted in 1 liter of warm water. Composition details, chemical-physical characteristics, and properties of the disinfected agent are summarized in the following table (Table 2);Cleaning: washing in an ultrasonic tank filled with the same solution as described above for 15 min, effectively promoting the breakdown and elimination of potential contaminants from HA surfaces;Rinsing: take the material and rinse it thoroughly under flowing water;Drying: the material must be thoroughly dried before being bagged;Packaging: dried HAs were packaged in a sterile barrier system (SBS) consisting of paper and polypropylene. The packaging included traceability data for the sterilization cycle, such as the sterilization operator’s signature, the autoclave-indicated sterilization cycle number, packaging date, and expiration date (set at 30 days from packaging). The packages must feature an external chemical indicator of class 1 (UNI EN ISO 11140-1), which indicates whether the package has undergone processing (process indicator) [30];Sterilization: in autoclave, cycle with bagged material at 134 °C [31,32];Storage (Figure 2).

### 2.3. Sample Sterility Evaluation

Extraction of the HA from the sterilization pack using sterile forceps under a laminar flow hood;Depositing the HA in a sterile 50 mL tube containing 10 mL Tryptone Soya Broth (TSB, Gibco, Fisher Scientific Italia, c/o Segreen Business Park, Milan, Italy) [33,34];Incubation of the HA for 24 h at 37 °C;Assessment for the presence of turbidity in the culture broth by an operator; if absent, the component is considered sterile.

### 2.4. Assessment of Surface Protein Levels

The concentration of residual protein on healing abutment was determined using a Micro BCA assay (Pierce Biotechnology, Rockford, Illinois).

The MicroBCA assay is based on the colorimetric change due to bicinchonic acid (BCA) chelating the copper ion (Cu^1+^) resulting from the reduction of copper ion (Cu^2+^) in the presence of protein and in an alkaline environment [35,36]. The colorimetric change goes from green to purple as the amount of protein in solution increases (Figure 3).

The color change is detected in the visible light spectrum using a spectrophotometer with an optical density (OD) of 562 nm.

In spectrophotometry, OD is the basis of quantitative chemical analysis and is linearly related to the concentration of the sample.

The steps applied to the HAs of each group were as follows:Each HA was removed from the bags using tweezers and deposited in sterile 1.5 milliliter (mL) tubes.Then, 500 µL of BCA solution was added to each test tube to completely cover each screw (Figure 4).Samples and blanks, i.e., sterile tubes without HAs and containing only BCS for the purpose of visual comparison, were subject to a 2 h incubation at 37 °C (Figure 5).Aliquots of 150 µL were added to 96-well plates in triplicate, and absorbance was read using a plate reader at 562 nm (Figure 6).Using known amounts of a reference protein (BSA, bovine serum albumin), a standard curve was created to trace the specific amount (ug/mL) of protein in each sample. Using the equation for the straight line, the amount of protein in ug/mL for each sample was obtained by substituting Y for the OD = 562 nm values obtained [37] (Figure 7).

### 2.5. Statistical Analysis

Statistical analysis was performed using specialized software (SPSS v20, Chicago, IL, USA). The normal distributions of the dependent variables were computed using Shapiro–Wilk tests. For assessment of surface protein levels, the significance of comparisons of means between groups was tested using one-way analysis of variance (ANOVA). Significance between groups was calculated using independent *t*-test. Tukey’s adjustment was computed for multiple comparisons. A *p*-value less than 0.05 was regarded as significant.

Statistical examination was conducted at a 95% significance level.

The determination of the study size was guided using a statistical power analysis, employing a *t*-test for two independent samples. We aimed for a significance level (α) of 0.05 and a power (1 − β) of 0.80, representing a commonly accepted balance between Type I and Type II errors. We calculated the required sample size for a two-sample *t*-test using the following formula:n = 2(σ2) (Zα/2 + Zβ)2/δ2
where σ is the estimated standard deviation; Zα/2 and Zβ are critical values for the chosen significance level and power, respectively; and delta is the effect size.

We conducted a sensitivity analysis on the effect size used in the power analysis. The required sample size for an effect size of 0.5 was determined to be 32. For an effect size of 0.8, the required sample size increased to 75.

## 3. Results

### 3.1. Sample Sterility Assessment

No turbidity was detected in the culture broth, proving the absence of bacterial growth in the examined sample. The decontamination and sterilization protocol were effective in removing any infective agents [38,39].

### 3.2. Surface-Attached Protein Levels

According to the results obtained with the MicroBCA assay, reused healing abutments, even after decontamination and sterilization, appear to be coated with surface proteins [40,41] (Figure 8).

The detection of the amount of protein on new healing abutments was used as reference for the residual protein detected on reused healing abutments.

The obtained results reveal a significant difference in the OD562 nm values between new and reused HA samples. The assay demonstrates how treated HAs were still contaminated by residual proteins (Figure 9).

After protein quantification using a BSA standard curve, the average residual protein concentration detected in the eluate from TG screws was 23.88 µg/mL (Figure 10).

Considering that the total amount of solution in which the screw was immersed is 500 µL, the average amount of protein present on each screw was almost 12 µg.

The sterilization procedure did not completely remove the proteins from the reused screws.

### 3.3. Statistical Results

Prior to statistical analyses, the normality of dependent variables was assessed using the Shapiro-Wilk test. Results indicated that all variables exhibited a non-significant departure from normality (*p* > 0.05), validating the use of parametric statistical techniques.

To explore differences in surface protein levels among various sample groups, a one-way ANOVA was conducted. This analysis revealed a significant variation among groups (F(2, 45) = 7.34, *p* < 0.01), indicating that at least one pair of groups significantly differed concerning detected protein quantity.

To identify specific differences in protein levels between new and reused healing abutments, an independent samples *t*-test was performed. Results showed a significant discrepancy between the two groups (t(30) = 4.52, *p* < 0.001), highlighting that the mean surface protein levels significantly differed between new and reused abutments.

To conduct multiple comparisons among groups, Tukey’s adjustment was applied. This procedure confirmed that differences between groups were statistically significant, indicating that each group significantly differed from the others (*p* < 0.05).

The mean quantity of surface proteins detected on reused healing abutments was 18.76 µg/mL (standard deviation = 3.62), whereas it was 9.35 µg/mL (standard deviation = 2.15) on new abutments. The mean difference in protein quantity between the two groups was 9.41 µg/mL (95% confidence interval [4.28, 14.54]).

## 4. Discussion

The primary objective of this study is to assess the effectiveness of a decontamination and sterilization protocol, one that could be practically implemented in several clinical settings, rendering healing abutments (HA) reasonably safe for reuse. To achieve this, the study adopts a comparative approach, scrutinizing two distinct groups: healing abutments that have been reused and subsequently subjected to sterilization versus brand-new healing abutments.

Upon evaluation of the sterility of our samples, a noteworthy observation emerged—the absence of any bacterial growth in either the test group (reused and sterilized HAs) or the control group (new HAs). This initial finding is encouraging, suggesting that the decontamination and sterilization protocol employed in this study may effectively mitigate bacterial contamination concerns associated with reused HA.

Consistent with our outcomes, a recent study sought to shed light on the efficacy of various sterilization approaches for HA. This study, examining 120 healing abutments, meticulously explored the impact of autoclave sterilization alone, autoclave sterilization combined with airflow polishing, and autoclave sterilization along with sodium hypochlorite treatment. Intriguingly, the results illuminated a significant accumulation of debris in the first two groups, while the third group displayed minimal staining. As a result, the authors confidently concluded that effective decontamination of reused HAs could indeed be achieved through specific sterilization procedures [42].

Extending the scope of our exploration, a recent systematic review conducted by Bidra and colleagues delved into the broader question of whether healing abutments and cover screws can be effectively re-sterilized for reuse. Their rigorous review of the existing literature identified a total of six pertinent studies. Remarkably, three of these studies arrived at the conclusion that routine cleaning methods alone were insufficient for fully decontaminating HA. In contrast, two studies incorporating an additional cleaning regimen successfully achieved complete removal of contaminants, while one study reported the complete elimination of debris from HA surfaces [43].

However, our investigation takes a deeper dive into the issue by examining the presence of protein residues. Our findings, derived from analysis, demonstrate that the sterilization process, while effective in many aspects, falls short in completely eradicating proteins from reused screws. This notable difference in protein content between new and reused screws underscores the persisting challenge of thorough decontamination and raises crucial questions about potential biological element transfer between patients.

Further substantiating these concerns, another study assessed 100 healing abutments sourced from eight different clinics. Even after undergoing sterilization, a staggering 99% of these HAs were found to be contaminated with proteins and peptides, raising alarming concerns regarding the potential transmission of biological elements from one patient to another [44].

Beginning with the study of Stacchi et al. [45], their investigation delved into a comparative evaluation of two distinct cleaning procedures applied to used HAs. The test HAs underwent a comprehensive decontamination journey, commencing with mechanical cleaning aided by disinfection sponges. This was followed by immersion in an ultrasonic bath and culminated with autoclaving, a rigorous regimen aimed at ensuring pristine cleanliness. However, intriguingly, despite these meticulous cleaning protocols, a significant discovery emerged. Out of a sample of 30 HAs in the test group, 11 were found to be contaminated. This unexpected finding underscores the challenges in achieving complete decontamination, even with the most stringent cleaning methodologies. Furthermore, in parallel research by the same team, the control HAs sourced from the same patients were subjected to a distinct automated cleaning system, again concluding with autoclaving. Astonishingly, all HAs within the control group were found to be contaminated. These revelations emphasize the complexity of the decontamination process and the need for further exploration into effective cleaning protocols.

Similarly, the inquiry by Sanchez-Garces et al. [46] ventured into the survival of microorganisms on 55 used and sterilized HAs. As anticipated, given the thorough sterilization procedures, no bacterial growth was detected on any HAs. However, the story did not end there. Significant amounts of debris were identified on the surfaces of these sterilized HAs, a phenomenon quantified through measurements of total organic carbon. This revelation underscores the intricate challenge of achieving a surface that is not only sterile but also free from residual debris, highlighting the multifaceted nature of the HA decontamination process.

Shifting the focus to in vivo investigations, Sennerby et al. [47] embarked on a study examining the biological response to used titanium screws harvested from human patients following replantation in rats. A notable increase in the number of activated macrophages and the development of fibrous encapsulation were observed around the used screws when compared to their pristine, never-used counterparts. While the most plausible explanation for this differential response points to a tissue reaction triggered by contaminants on the surface of the used screws, the precise nature of these differences remains a captivating mystery, one that calls for further exploration and investigation into the intricacies of the biological response to reused HAs.

Adding another layer to this multifaceted exploration, Cakan et al. [48] conducted an examination of 60 used but sterilized HAs collected from six different commercial suppliers. In a macroscopic analysis of these HAs, intriguing findings emerged. Contaminated screw grooves were identified in 10.5% of the samples, revealing the challenges in achieving pristine cleanliness in the often intricate and recessed areas of HAs. Additionally, a noteworthy 5.2% of these HAs had their driver slots filled with debris, further underscoring the complexity of the decontamination process and the potential areas where contaminants can persist.

## 5. Conclusions

From an infectious point of view, considering the higher resistance of bacteria compared to viral agents, it can be stated that the reuse procedure could be considered safe. As far as the presence of protein components is concerned, the sterilization process does not completely remove proteins from the reused screws; however, the clinical significance of the difference recorded between the control and test group cannot be determined a priori. Based on these considerations, further studies should be conducted by applying other sterilization procedures to strongly reduce the detectable protein amount and by evaluating the possible biomechanical implications of this type of treatment on the healing screws.

## Figures and Tables

**Figure 1 bioengineering-11-00387-f001:**
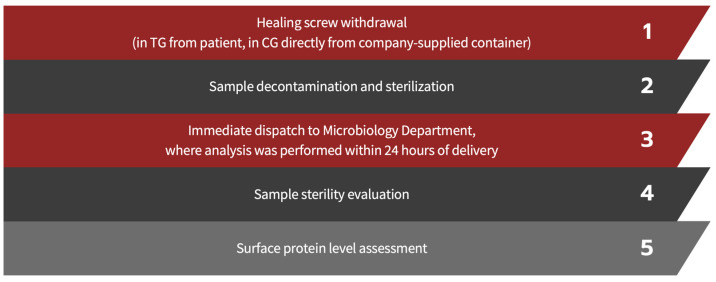
Workflow details.

**Figure 2 bioengineering-11-00387-f002:**
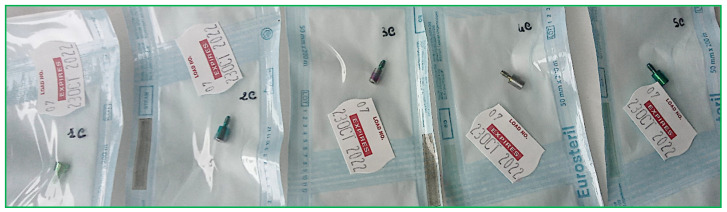
HAs following the above-described decontamination and sterilization process.

**Figure 3 bioengineering-11-00387-f003:**
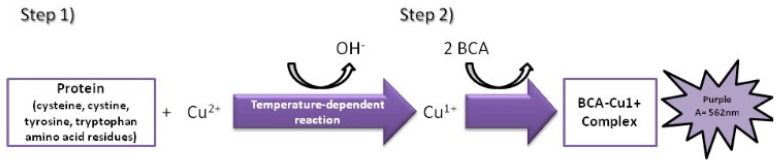
Colorimetric variation based on reduction of Cu^2+^ in the presence of protein and in an alkaline environment and subsequent chelation reaction between BCA and Cu^1+^ resulting from the first reaction.

**Figure 4 bioengineering-11-00387-f004:**
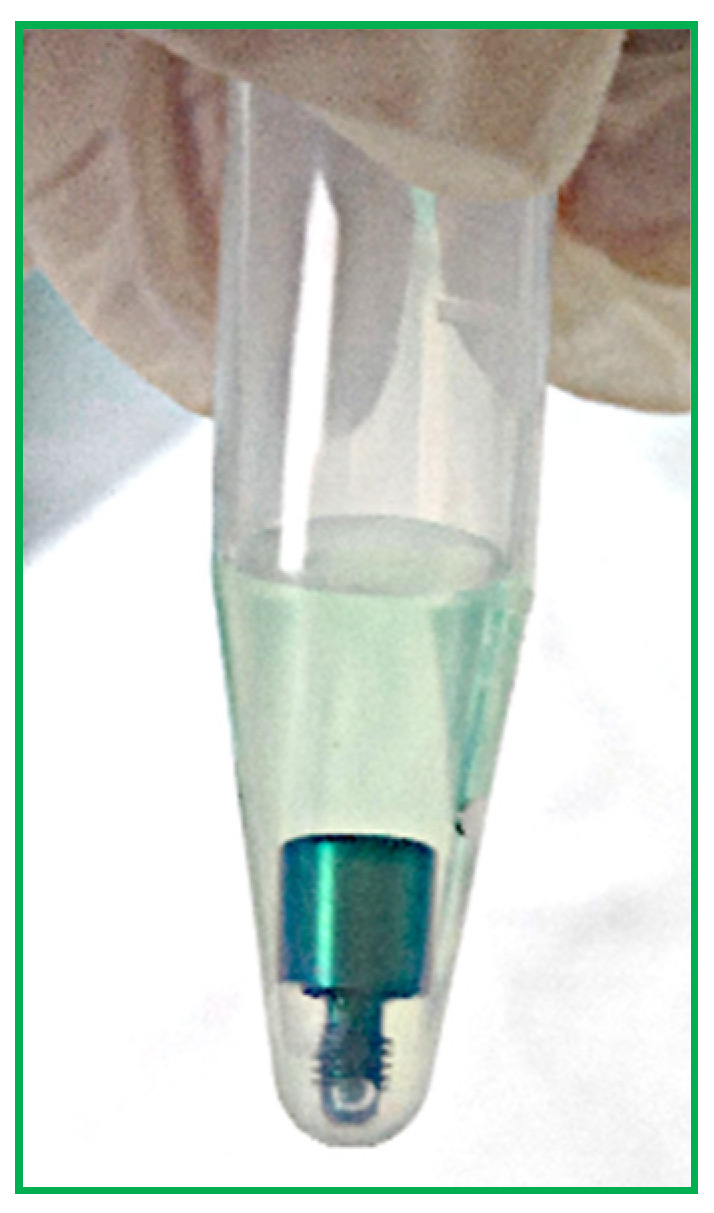
Sterile tube containing a HA in BCA solution.

**Figure 5 bioengineering-11-00387-f005:**
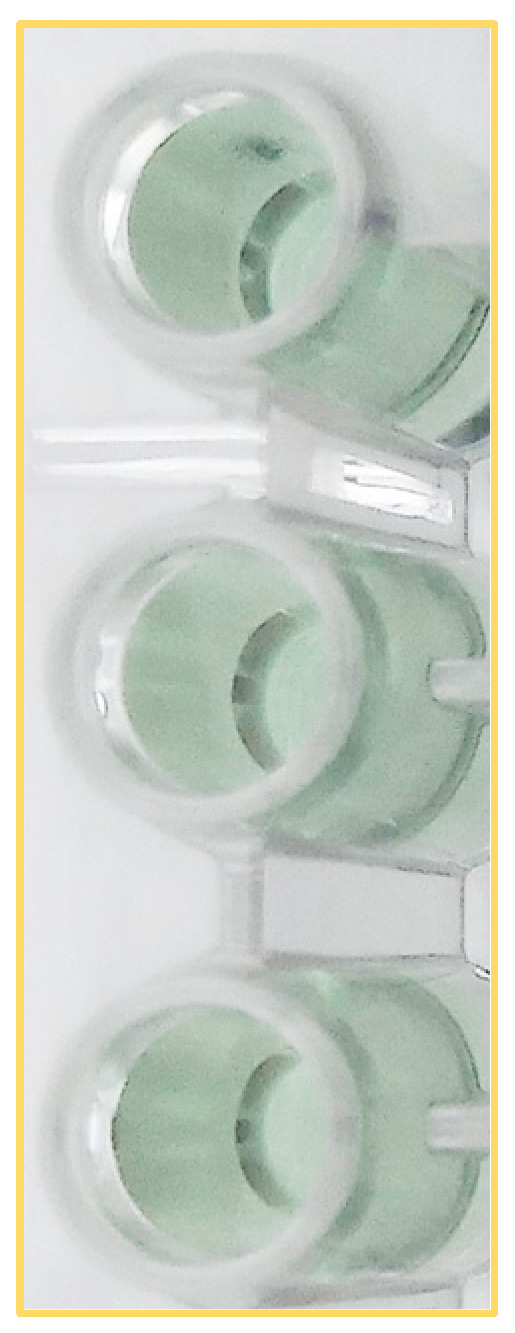
Sterile tubes free of HAs and containing only BCS.

**Figure 6 bioengineering-11-00387-f006:**
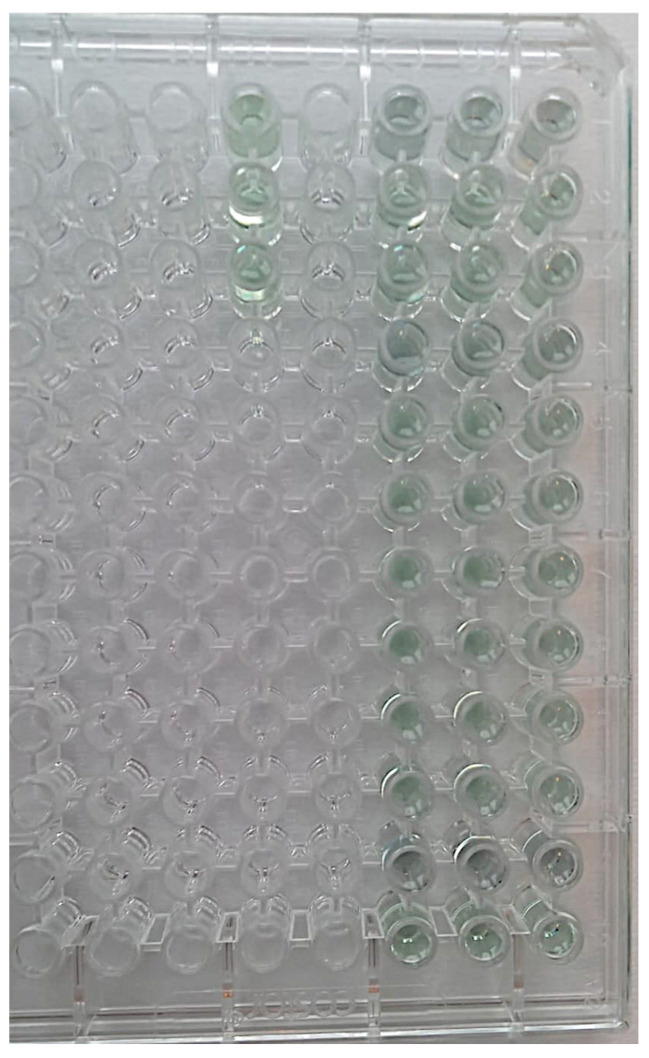
Aliquots of 150 µL were added to 96-well plates in triplicate. All tubes retain the green coloring as the reaction to test for surface proteins has not yet been performed. The three sterile tubes free of HAs and containing only BCS are placed apart on the left to distinguish them from the sample under examination.

**Figure 7 bioengineering-11-00387-f007:**
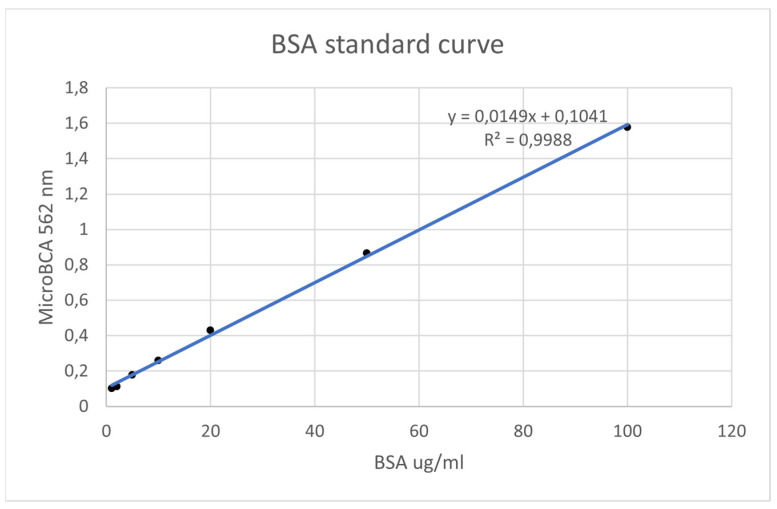
BSA standard curve.

**Figure 8 bioengineering-11-00387-f008:**
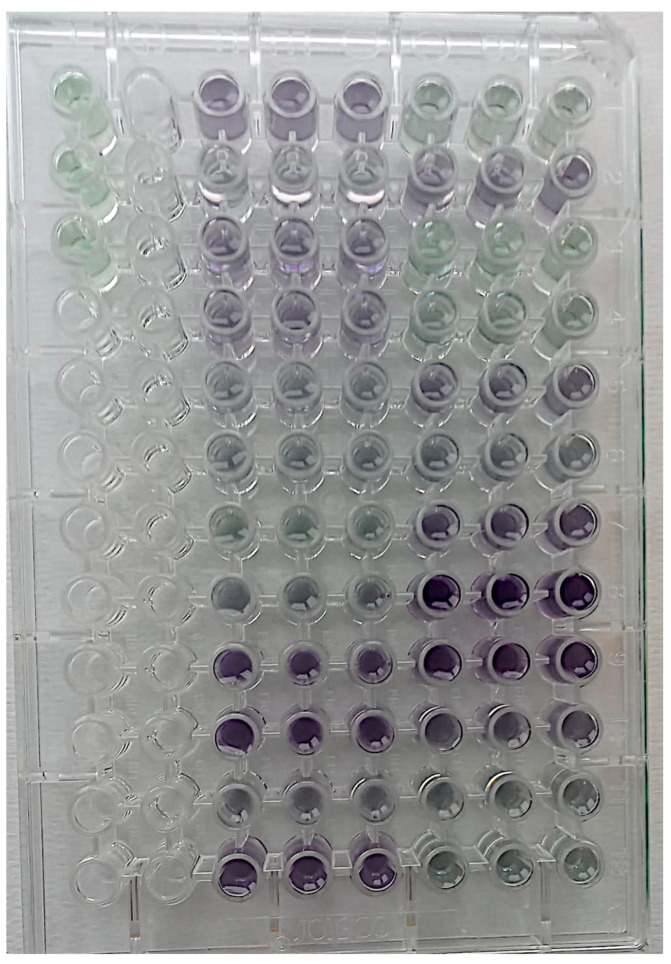
Configuration obtained with the MicroBCA assay. Reused healing screws with surface proteins appear violet in color.

**Figure 9 bioengineering-11-00387-f009:**
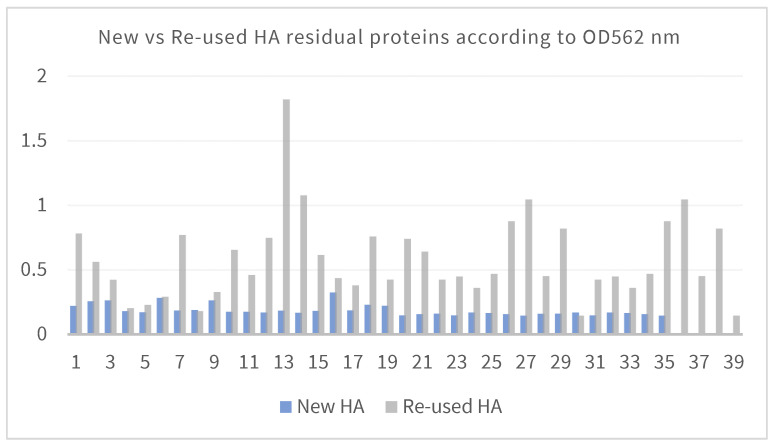
Difference in the OD562 nm values between new and reused HA samples.

**Figure 10 bioengineering-11-00387-f010:**
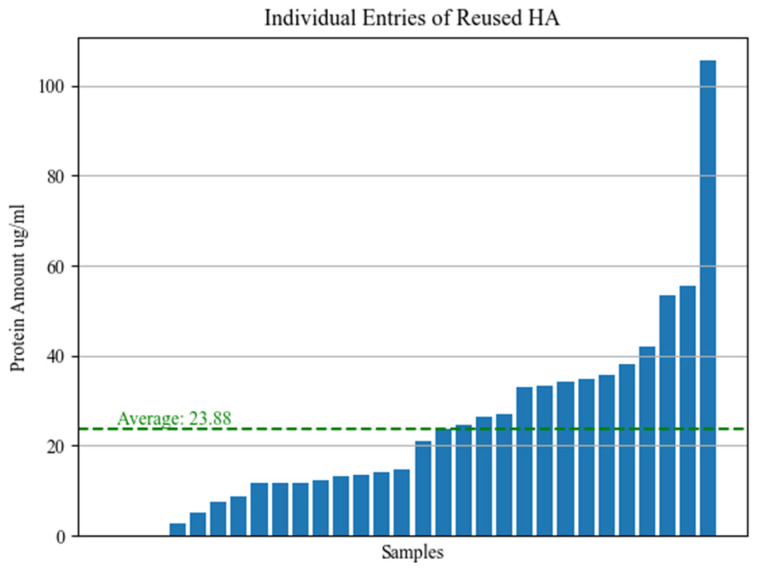
Average residual protein concentration detected in the eluate from TG screws.

**Table 1 bioengineering-11-00387-t001:** Details of type and features of selected healing screws.

**Morphology**	Cylindrical screw geometry with cylindrical head for screwed prosthesis	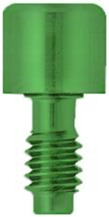
**Diameter**	3.8 mm	
**Height**	3 mm	
**Material**	Titanium alloy Ti6Al4V grade 5 ELI ref. standard ASTM F136-13	
**Sterilization**	Non-sterile, autoclavable	
**CE classification**	Class IIb, rule 8	

**Table 2 bioengineering-11-00387-t002:** Disinfecting solution composition and properties.

** Composition **		
Active ingredients	Sodium percarbonate	20%
	Tetraacetylethylenediamine	15%
Excipients	Washing agents, non-ionic surfactants, stabilizers and co-formulants	
** Chemical-physical characteristics **		
Appearance	White to yellowish granulate	
Specific weight	1 g/mL	
pH	10	
Flash point	>100 °C	
Storage temperature	0 °C < T < 25 °C	
** Properties **	Concentration	Time of contact
Mycobactericidal activity	2%	15 min
Sporicidal activity	2%	30 min
Virucidal activity	2%	15 min

## Data Availability

All data can be seen in this manuscript.

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
