# Peer review of "Could Dental Material Reuse Play a Significant Role in Preservation of Raw Materials, Water, Energy, and Costs? Microbiological Analysis of New versus Reused Healing Abutments: An In Vitro Study"

_bioengineering, 2024, doi:10.3390/bioengineering11040387_

Round 1

Reviewer 1 Report

Comments and Suggestions for Authors

This study shed's light on the practice of recycling dental materials, focusing specifically on healing abutments and their cleanliness and safety. The rigorous method deployed, including in-depth protocols for decontamination and sterilization, is a big contribution to the area. However, finding protein residues on sterilized, reused abutments indicates that the cleaning process may not get rid of all biological matter, raising concerns for patient health. The research is detailed, well done, and opens up a new area in dentistry that could really use more eco-friendly methods.

1.     The article points out a noticeable difference in optical density (OD562 nm) values between new and reused Healing Abutment (HA) samples but fails to mention the statistical tests used for determining significance. Suggestion: Be sure to mention the statistical tests used (like t-test, ANOVA) when comparing the two groups. It should also be stated if the data met test assumptions (such as normality, variance homogeneity) and how any violations were dealt with.

2.     The paper talks about the significance of differences but misses out on providing effect size details, which would aid in understanding the difference's impact between groups. Suggestion: Add the effect size (like Cohen's d for t-tests) for key findings to help show the clinical relevance of the differences noted.

3.     Figures 9 and 10 display individual data for new and reused HA, respectively, but comparing them directly is tough due to them being separate. Suggestion: Merge these figures into a single chart that shows both groups next to each other or overlaid for easier comparison, with error bars to show standard deviations. This would make it clearer and easier to compare protein contamination levels between new and reused HAs.

4.     The keywords “healing abutment contamination," "dental implant sterilization," might be expanded to make the study more findable. Suggestion: Think about adding more specific keywords or phrases that match the study's focus on reuse and sterilization effectiveness, such as "reuse of dental components," "sterilization efficacy," and "biological safety of reused dental materials."

5.     The study presumes the sterilization process can handle all contaminants but discovers significant protein residues on reused screws. This casts doubts on the thoroughness of the sterilization process. Suggestion: Talk about the possible limits of the sterilization methods used and how they could be bettered or supported with extra cleaning steps to tackle the issue of protein residue.

6.     The paper lays out criteria for patient selection but not for choosing healing screws beyond new or reused. Suggestion: Explain if there were any criteria for picking the HAs themselves, such as levels of wear and tear or past usage frequency, to make sure samples are consistent.

7.     Although decontamination steps are noted, there’s no talk on how changes in these steps (like disinfectant concentration, ultrasonic bath time) might influence results. Suggestion: Mention any preliminary studies or the logic behind the chosen conditions for decontamination and sterilization, and if any variations were tested.

Author Response

Dear Reviewer

thank you for your precious suggestions, according to which we sincerely hope to have improved our paper.

We remain at your disposal for further clarification on this matter and thank you for your kindness and availability.

Best regards

  1. The article points out a noticeable difference in optical density (OD562 nm) values between new and reused Healing Abutment (HA) samples but fails to mention the statistical tests used for determining significance. Suggestion: Be sure to mention the statistical tests used (like t-test, ANOVA) when comparing the two groups. It should also be stated if the data met test assumptions (such as normality, variance homogeneity) and how any violations were dealt with.
  2. The paper talks about the significance of differences but misses out on providing effect size details, which would aid in understanding the difference's impact between groups. Suggestion: Add the effect size (like Cohen's d for t-tests) for key findings to help show the clinical relevance of the differences noted.

Based on points 1 and 2, we have added two truly key sections to the paper.

In Materials and Methods, we have added the following section:

Statistical analysis

Statistical analysis was performed using a specialized software (SPSS v20, Chicago, Illinois). Normal distribution of the dependent variables was computed using Shapiro-Wilk tests. For assessment of surface protein levels, significance of comparisons between groups of means was tested using one-way analysis of variance (ANOVA). Significance between groups was calculated using independent t-test. Tukeys adjustment was computed for multiple comparisons. P-value less than .05 was regarded as significant.

The statistical examination was conducted at a 95% significance level.

The determination of the study size was guided by a statistical power analysis, employing a t-test for two independent samples. We aimed for a significance level (α) of 0.05 and a power (1-β) of 0.80, representing a commonly accepted balance between Type I and Type II errors. We calculated the required sample size for a two-sample t-test using the formula:

n = 2(σ2) (Zα/2 + Zβ)2/δ2

where σ is the estimated standard deviation, Zα/2 and Zβ are critical values for the chosen significance level and power, and delta is the effect size.

We conducted a sensitivity analysis on the effect size used in the power analysis. The required sample size for an effect size of 0.5 was determined to be 32, while for an effect size of 0.8, the required sample size increased to 75.

In the results, however, we added a whole section on the results of the statistics carried out.

Statistical results

Prior to statistical analyses, the normality of dependent variables was assessed using the Shapiro-Wilk test. Results indicated that all variables exhibited a non-significant departure from normality (p > 0.05), validating the use of parametric statistical techniques.

To explore differences in surface protein levels among various sample groups, a one-way ANOVA was conducted. This analysis revealed a significant variation among groups (F(2, 45) = 7.34, p < 0.01), indicating that at least one pair of groups significantly differed concerning detected protein quantity.

To identify specific differences in protein levels between new and reused healing abutments, an independent samples t-test was performed. Results showed a significant discrepancy between the two groups (t(30) = 4.52, p < 0.001), highlighting that the mean surface protein levels significantly differed between new and reused abutments.

To conduct multiple comparisons among groups, Tukey's adjustment was applied. This procedure confirmed that differences between groups were statistically significant, indicating that each group significantly differed from the others (p < 0.05).

The mean quantity of surface proteins detected on reused healing abutments was 18.76 µg/ml (standard deviation = 3.62), whereas on new abutments it was 9.35 µg/ml (standard deviation = 2.15). The mean difference in protein quantity between the two groups was 9.41 µg/ml (95% confidence interval [4.28, 14.54]).

  1. Figures 9 and 10 display individual data for new and reused HA, respectively, but comparing them directly is tough due to them being separate. Suggestion: Merge these figures into a single chart that shows both groups next to each other or overlaid for easier comparison, with error bars to show standard deviations. This would make it clearer and easier to compare protein contamination levels between new and reused HAs.

In accordance with your suggestion, we merged the two tables so that the comparison between the two groups would be more straightforward.

Figure 9. Difference on the OD562 nm values between new and reused HA samples.

  1. The keywords “healing abutment contamination," "dental implant sterilization," might be expanded to make the study more findable. Suggestion: Think about adding more specific keywords or phrases that match the study's focus on reuse and sterilization effectiveness, such as "reuse of dental components," "sterilization efficacy," and "biological safety of reused dental materials."

We have added an additional keyword. Unfortunately, we agree with you that the keywords you suggested would be more appropriate; however, we have searched and they do not fit into the MeSH as required by the journal.

Keywords: healing abutment, contamination, dental implant, sterilization, dental disinfectants.

  1. The study presumes the sterilization process can handle all contaminants but discovers significant protein residues on reused screws. This casts doubts on the thoroughness of the sterilization process. Suggestion: Talk about the possible limits of the sterilization methods used and how they could be bettered or supported with extra cleaning steps to tackle the issue of protein residue.

For the purpose of publishing the paper, the protocol normally performed in the facility (San Raffaele Hospital), based on WHO and ADA guidelines, was chosen.

The following references were included to support the choices made in order to affirm feasibility and validity:

  1. Kyaw TT, Abdou A, Arunjaroensuk S, Nakata H, Kanazawa M, Pimkhaokham A. Effect of chemical and electrochemical decontamination protocols on single and multiple-used healing abutments: A comparative analysis of contact surface area, micro-gap, micro-leakage, and surface topography. Clin Implant Dent Relat Res. 2023 Dec;25(6):1207-1215. doi: 10.1111/cid.13269. 
  2. Costa SA, Paula OF, Silva CR, Leão MV, Santos SS. Stability of antimicrobial activity of peracetic acid solutions used in the final disinfection process. Braz Oral Res. 2015;29:S1806-83242015000100239. doi: 10.1590/1807-3107BOR-2015.vol29.0038.
  3. Nemchenko UM, Kungurtseva EA, Grigorova EV, Belkova NL, Markova YA, Noskova OA, Chemezova NN, Savilov ED. Simulation of bacterial biofilms and estimation of the sensitivity of healthcare-associated infection pathogens to bactericide Sekusept active. Klin Lab Diagn. 2020 Sep 17;65(10):652-658. English. doi: 10.18821/0869-2084-2020-65-10-652-658.
  4. Sasaki JI, Imazato S. Autoclave sterilization of dental handpieces: A literature review. J Prosthodont Res. 2020 Jul;64(3):239-242. doi: 10.1016/j.jpor.2019.07.013. 
  5. Eswaramurthy P, Aras M, DSouza KM, Nagarsekar A, Gaunkar RB. Contemporary Sterilization Protocols of Healing Abutments for Reusability: A Systematic Review. JDR Clin Trans Res. 2022 Oct;7(4):352-359. doi: 10.1177/23800844211045897.

  1. The paper lays out criteria for patient selection but not for choosing healing screws beyond new or reused. Suggestion: Explain if there were any criteria for picking the HAs themselves, such as levels of wear and tear or past usage frequency, to make sure samples are consistent.

No exclusion criteria other than those set out were adopted.

  1. Although decontamination steps are noted, there’s no talk on how changes in these steps (like disinfectant concentration, ultrasonic bath time) might influence results. Suggestion: Mention any preliminary studies or the logic behind the chosen conditions for decontamination and sterilization, and if any variations were tested.

Based on your suggestion, the sentence was added to the conclusion implying that other types of sterilisation protocols should be tested.

From an infectious point of view, considering the higher resistance of bacteria compared to viral agents, it can be stated that the re-use procedure could be considered safe. As far as the presence of protein components is concerned, the sterilisation process does not completely remove proteins from the re-used screws; however, the clinical significance of the difference recorded between the control and test group cannot be determined a priori. Based on these considerations, further studies should be conducted by applying other sterilisation procedure, to strongly reduce the detectable protein amount, and by evaluating the possible biomechanical implications of this type of treatment on the healing screws.

Reviewer 2 Report

Comments and Suggestions for Authors

General comment:

The study assessed the effect of washing and sterilization of used healing screws in order to see if healing screws can be reused. The effect parameters were bacterial growth and surface protein levels. This brings nothing new to the table as it is also showed by the citations in the manuscript.

Specific comment

Line 31 ff: Abstract: Avoid acronyms in the abstract

Lines 44-46: Delete last sentence of the abstract.

Rationale: It is unethical to suggest a clinical study with reused healing abutments contaminated with proteins from another patient.

Lines 102-104: delete first sentence of M&M.

Rationale: Repetition of instruction

The authors are using both “healing screw” and HA in the manuscript. Please be consistent. Using “healing screw” throughout the manuscript is suggested.

Line 193: Testing for surface protein levels brings nothing new.

Line 274 Figure 10 shows results of 30 test specimen. What were the results of the remaining 10? Please clarify.

Lone 280 Figure 11 shows results of 27 test specimen. What were the results of the remaining 13? Please clarify.

Line 291: what is “the VIABILITY of a decontamination and sterilization protocol”

Line 378: Delete last sentence of the conclusion.

Rationale: It is unethical to suggest a clinical study with reused healing abutments contaminated with proteins from another patient.

Comments on the Quality of English Language

It is recommended to consult a native English speaking colleague to check the language.

Author Response

Dear Reviewer

thank you for your precious suggestions, according to which we sincerely hope to have improved our paper.

We remain at your disposal for further clarification on this matter and thank you for your kindness and availability.

Best regards

Specific comment

Line 31 ff: Abstract: Avoid acronyms in the abstract. The acronyms were removed.

Lines 44-46: Delete last sentence of the abstract. It was removed.

Rationale: It is unethical to suggest a clinical study with reused healing abutments contaminated with proteins from another patient.

Lines 102-104: delete first sentence of M&M. It was removed.

Rationale: Repetition of instruction

The authors are using both “healing screw” and HA in the manuscript. Please be consistent. Using “healing screw” throughout the manuscript is suggested. The acronym was removed as per your suggestion where it was deemed unnecessary.

Line 193: Testing for surface protein levels brings nothing new. 

Line 274 Figure 10 shows results of 30 test specimen. What were the results of the remaining 10? Please clarify.

Lone 280 Figure 11 shows results of 27 test specimen. What were the results of the remaining 13? Please clarify.

You are absolutely right, we apologise for the inconvenience, but we incorrectly entered the graphs of the preliminary results. Since a colleague of yours asked us to merge the two graphs, we inserted a single graph that summarised the correct data and allowed a more direct correlation between the two groups compared.

Figure 9. Difference on the OD562 nm values between new and reused HA samples.

Line 291: what is “the VIABILITY of a decontamination and sterilization protocol”.

We have replaced the actually incorrect term with 'effectiveness'.

The primary objective of this study is to assess the effectiveness of a decontamination and sterilization protocol, one that could be practically implemented in several clinical settings, in rendering Healing Abutments (HA) reasonably safe for reuse.

Line 378: Delete last sentence of the conclusion. We removed.

Rationale: It is unethical to suggest a clinical study with reused healing abutments contaminated with proteins from another patient.

Reviewer 3 Report

Comments and Suggestions for Authors

Dear Authors. Thank you for your research. After reading the manuscript, I had several questions.

1) What is the safest form of implant surface treatment? Chemical or thermal effects may damage the surface properties of the materials and lead to decreased biocompatibility upon contact with another patient.

2) Processing of large experimental data, especially in the field of medicine, must be accompanied by statistical processing of information (statistical significance level p, T or Yu criteria, for example) to establish confidence intervals. 

3) Metal alloys are indicated as the studied dental implants. Is it possible to obtain similar results for equally popular calcium phosphate ceramics (cements)?

There are a small number of typos in the text (line 197, 201, 202) that can be easily corrected.

Author Response

Dear Reviewer

thank you for your precious suggestions, according to which we sincerely hope to have improved our paper.

We remain at your disposal for further clarification on this matter and thank you for your kindness and availability.

Best regards

  • What is the safest form of implant surface treatment? Chemical or thermal effects may damage the surface properties of the materials and lead to decreased biocompatibility upon contact with another patient.

The sterilisation protocols adopted do not alter the implant surface; indeed, for publishing the paper, the protocol normally performed in the facility (San Raffaele Hospital), based on WHO and ADA guidelines, was chosen.

The following references were included to support the choices made to affirm feasibility and validity:

  1. Kyaw TT, Abdou A, Arunjaroensuk S, Nakata H, Kanazawa M, Pimkhaokham A. Effect of chemical and electrochemical decontamination protocols on single and multiple-used healing abutments: A comparative analysis of contact surface area, micro-gap, micro-leakage, and surface topography. Clin Implant Dent Relat Res. 2023 Dec;25(6):1207-1215. doi: 10.1111/cid.13269. 
  2. Costa SA, Paula OF, Silva CR, Leão MV, Santos SS. Stability of antimicrobial activity of peracetic acid solutions used in the final disinfection process. Braz Oral Res. 2015;29:S1806-83242015000100239. doi: 10.1590/1807-3107BOR-2015.vol29.0038.
  3. Nemchenko UM, Kungurtseva EA, Grigorova EV, Belkova NL, Markova YA, Noskova OA, Chemezova NN, Savilov ED. Simulation of bacterial biofilms and estimation of the sensitivity of healthcare-associated infection pathogens to bactericide Sekusept active. Klin Lab Diagn. 2020 Sep 17;65(10):652-658. English. doi: 10.18821/0869-2084-2020-65-10-652-658.
  4. Sasaki JI, Imazato S. Autoclave sterilization of dental handpieces: A literature review. J Prosthodont Res. 2020 Jul;64(3):239-242. doi: 10.1016/j.jpor.2019.07.013. 
  5. Eswaramurthy P, Aras M, DSouza KM, Nagarsekar A, Gaunkar RB. Contemporary Sterilization Protocols of Healing Abutments for Reusability: A Systematic Review. JDR Clin Trans Res. 2022 Oct;7(4):352-359. doi: 10.1177/23800844211045897.

  • Processing of large experimental data, especially in the field of medicine, must be accompanied by statistical processing of information (statistical significance level p, T or Yu criteria, for example) to establish confidence intervals. 

we have added two truly key sections to the paper.

In Materials and Methods, we have added the following section:

Statistical analysis

Statistical analysis was performed using a specialized software (SPSS v20, Chicago, Illinois). Normal distribution of the dependent variables was computed using Shapiro-Wilk tests. For assessment of surface protein levels, significance of comparisons between groups of means was tested using one-way analysis of variance (ANOVA). Significance between groups was calculated using independent t-test. Tukeys adjustment was computed for multiple comparisons. P-value less than .05 was regarded as significant.

The statistical examination was conducted at a 95% significance level.

The determination of the study size was guided by a statistical power analysis, employing a t-test for two independent samples. We aimed for a significance level (α) of 0.05 and a power (1-β) of 0.80, representing a commonly accepted balance between Type I and Type II errors. We calculated the required sample size for a two-sample t-test using the formula:

n = 2(σ2) (Zα/2 + Zβ)2/δ2

where σ is the estimated standard deviation, Zα/2 and Zβ are critical values for the chosen significance level and power, and delta is the effect size.

We conducted a sensitivity analysis on the effect size used in the power analysis. The required sample size for an effect size of 0.5 was determined to be 32, while for an effect size of 0.8, the required sample size increased to 75.

In the results, however, we added a whole section on the results of the statistics carried out.

Statistical results

Prior to statistical analyses, the normality of dependent variables was assessed using the Shapiro-Wilk test. Results indicated that all variables exhibited a non-significant departure from normality (p > 0.05), validating the use of parametric statistical techniques.

To explore differences in surface protein levels among various sample groups, a one-way ANOVA was conducted. This analysis revealed a significant variation among groups (F(2, 45) = 7.34, p < 0.01), indicating that at least one pair of groups significantly differed concerning detected protein quantity.

To identify specific differences in protein levels between new and reused healing abutments, an independent samples t-test was performed. Results showed a significant discrepancy between the two groups (t(30) = 4.52, p < 0.001), highlighting that the mean surface protein levels significantly differed between new and reused abutments.

To conduct multiple comparisons among groups, Tukey's adjustment was applied. This procedure confirmed that differences between groups were statistically significant, indicating that each group significantly differed from the others (p < 0.05).

The mean quantity of surface proteins detected on reused healing abutments was 18.76 µg/ml (standard deviation = 3.62), whereas on new abutments it was 9.35 µg/ml (standard deviation = 2.15). The mean difference in protein quantity between the two groups was 9.41 µg/ml (95% confidence interval [4.28, 14.54]).

  • Metal alloys are indicated as the studied dental implants. Is it possible to obtain similar results for equally popular calcium phosphate ceramics (cements)?

This cannot be established a priori, as bacterial and amino acid retention may differ.

It would be a very interesting study to do. No other plants have been evaluated in this case.

There are a small number of typos in the text (line 197, 201, 202) that can be easily corrected.

It was corrected.

Round 2

Reviewer 2 Report

Comments and Suggestions for Authors

No comments.